# Mindfulness-Based Stress Reduction for Systemic Lupus Erythematosus: A Mixed-Methods Pilot Randomized Controlled Trial of an Adapted Protocol

**DOI:** 10.3390/jcm10194450

**Published:** 2021-09-28

**Authors:** Renen Taub, Danny Horesh, Noa Rubin, Ittai Glick, Orit Reem, Gitit Shriqui, Nancy Agmon-Levin

**Affiliations:** 1Department of Psychology, Bar-Ilan University, Ramat Gan 5290002, Israel; danny.horesh@biu.ac.il (D.H.); badashnoa@gmail.com (N.R.); 2Department of Psychiatry, New York University School of Medicine, 1 Park Ave., New York, NY 10016, USA; 3Shachaf Clinic for Stress Reduction, Chaim Sheba Medical Center, Tel-Hashomer, Ramat Gan 52621, Israel; ittai.glick@gmail.com (I.G.); oritreem@gmail.com (O.R.); 4Psychodharma, 6 George Wise Street, Tel Aviv 6997705, Israel; gitit11@gmail.com; 5Clinical Immunology, Angioedema and Allergy, Center for Autoimmune Diseases, Chaim Sheba Medical Center, Tel-Hashomer, Ramat Gan 52621, Israel; nancy.Agmon-Levin@sheba.health.gov.il

**Keywords:** systemic lupus erythematosus (SLE), autoimmune diseases, mindfulness-based stress reduction (MBSR), mindfulness, psychotherapy

## Abstract

Background: The psychological effects of systemic lupus erythematosus (SLE) are tremendous. This pilot mixed-methods randomized controlled trial aimed to evaluate the effects of a mindfulness-based stress reduction (MBSR) adapted protocol on psychological distress among SLE patients. Methods: 26 SLE patients were randomly assigned to MBSR group therapy (*n* = 15) or a waitlist (WL) group (*n* = 11). An adapted MBSR protocol for SLE was employed. Three measurements were conducted: pre-intervention, post-intervention and 6-months follow up. A sub-sample (*n* = 12) also underwent qualitative interviews to assess their subjective experience of MBSR. Results: Compared to the WL, the MBSR group showed greater improvements in quality of life, psychological inflexibility in pain and SLE-related shame. Analysis among MBSR participants showed additional improvements in SLE symptoms and illness perception. Improvements in psychological inflexibility in pain and SLE-related shame remained stable over six months, and depression levels declined steadily from pre-treatment to follow-up. Qualitative analysis showed improvements in mindfulness components (e.g., less impulsivity, higher acceptance), as well as reduced stress following MBSR. Conclusions: These results reveal the significant therapeutic potential of MBSR for SLE patients. With its emphasis on acceptance of negative physical and emotional states, mindfulness practice is a promising treatment option for SLE, which needs to be further applied and studied.

## 1. Introduction

Systemic Lupus Erythematosus (SLE) is an autoimmune disease, the intensity of which varies between mild and severe. Apart from classical criteria such as joint pain, arthritis, skin lesions, and photosensitivity, it involves various organ-specific manifestations, as well as chronic fatigue and reduced quality of life [1,2]. The prevalence of SLE vary by age and gender, as it predominantly affects women in their 20′s and 30′s [3]. Furthermore, SLE is considered a stress-related disease, and in many cases symptoms are worsened under stressful conditions [4,5].

In their model of SLE symptom types, Pisetsky et al. [6] have coined the term “Type 2” symptoms. These include symptoms such as fatigue, widespread body pain, depression, anxiety, cognitive dysfunction, and sleep disturbance. Such symptoms usually do not respond to therapy with immunosuppression or corticosteroids, even with escalation of doses. Neurological and psychiatric manifestations affect the majority of SLE patients, and several of these manifestations define a disease criterion [7]. Throughout their lives, 65% of patients with SLE are diagnosed with a mood or anxiety disorder, including major depression (47%), specific phobia (24%), panic disorder (16%), obsessive-compulsive disorder (9%), and bipolar disorder (6%). Thus, psychological effects related to SLE are significant, and exert a considerable impact on patients’ quality of life [8]. SLE patients also experience cognitive impairment in various fields, and disease activity is correlated with cognitive dysfunction, along with self-reported fatigue, pain, and negative mood [9]. Type 2 symptoms of SLE are today recognized as a highly important aspect of the disease, which has yet to receive sufficient clinical attention [10]. Thus, despite these significant psychological difficulties experienced by individuals with SLE, there is a scarcity of research examining novel psychotherapeutic interventions in this field [11]. In this study, we focus on mindfulness-based therapy and its effects on different aspects of SLE.

Mindfulness involves “paying attention in a particular way: on purpose, in the present moment, and non-judgmentally” [12]. It refers to the cultivation of conscious awareness and attention on a moment-to-moment basis, with an open and non-judgmental attitude [13]. According to a growing body of evidence, mindfulness-based interventions (MBI) may improve coping with pain and psychological symptoms [14,15]. The most well-known variant of MBI is called Mindfulness–Based Stress Reduction (MBSR), a group therapy program that provides systemic training in mindfulness meditation as a self-regulation approach to stress reduction [16].

In its original version, MBSR is an eight-week program in mindfulness training. The standard program includes weekly group sessions of 2–2.5 h and one full-day session after six to seven weeks. The program’s core elements consist of various mental and physical mindfulness exercises: (1) body-scan exercises (paying close attention to all body parts, from head to toe), (2) mental exercises focusing one’s attention on breathing, (3) physical exercises (e.g., walking meditation) focusing on being aware of bodily sensations and one’s own limits during exercises, and (4) practicing being fully aware during everyday activities. Essential to all parts of the program is developing an accepting and non-reactive attitude to what one experiences in the present moment. In each session, time is allocated for group members to reflect together on what they experience when they practice mindfulness. Between sessions, participants are instructed to listen to 30–45 min guided exercises in body-scan, sitting meditation, focusing on breathing and yoga stretching [17].

Surprisingly, despite its well-documented effects on rheumatic and autoimmune diseases [18], research on MBI among SLE has so far been very scarce, and showed notable methodological limitations. To date, there has been only one randomized controlled trial of MBSR among SLE patients, focusing on a very limited set of outcome measures, none of which included physical or psychiatric symptoms [19]. Two other previous studies applied mindfulness-based cognitive therapy (MBCT) on SLE patients [20,21,22]. MBCT is another variant of MBI, which was originally designed to treat depression [23], and thus is less focused on stress reduction per-se. Here too, outcome measures were quite limited in scope. Finally, a recent study applied a mindfulness protocol specifically focused on meta-cognition (i.e., one’s ability to “think about one’s thoughts”) among individuals with SLE, showing improvements in psychological well-being, with no follow-up long-term assessment [24].

Furthermore, due to SLE patients’ physical limitations, administering the generic MBSR intervention may prove as both challenging and ineffective, as individuals may feel too sick to participate or experience difficulties with prolonged sitting/meditating, which may, in turn, entail high dropout rates [25]. Thus, a newly adapted protocol is called for. Finally, in order to achieve a fuller understanding of patients’ experiences in therapy, mixed-methods studies are needed, combining both quantitative and qualitative research methods.

In this study, we aim to fill these large gaps in research and clinical work, by conducting a randomized controlled trial of MBSR for SLE patients. The aims of this study were to evaluate the effects of MBSR on various psychological (i.e., “Type 2” symptoms) and physical outcomes, including reported SLE symptoms, health related quality of life, major depression and psychological inflexibility in pain, as well as SLE-related shame and illness perception. Due to the wide heterogeneity of SLE symptoms, we will also specifically assess weather changes in specific SLE symptoms following MBSR were associated with changes in psychological measures, in order to understand weather physical and mental health processes work together or separately in SLE psychotherapy. For this study, a unique MBSR protocol was constructed, to meet the specific needs of SLE patients. The study employs a mixed-methods approach, which has never been employed in SLE mindfulness research.

## 2. Materials and Methods

### 2.1. Participants and Procedure

SLE patients (Mean age = 41.62, SD = 11.78, Range: 22–64) were recruited from two major sources: 1. The Center for Autoimmune Diseases at the Sheba Medical Center in Israel. 2. Ads posted on social media (online groups and forums of SLE patients), as well as sent to organizations supporting SLE patients. A total of 85 individuals showed initial interest in the study. The first phone call presented patients with information about the intervention, and also included an initial screening-out of 26 participants, due to lack of interest after hearing about the intervention, inability to participate due to the place or time of the intervention, and lack of formal SLE diagnosis. The remaining 59 patients underwent in-depth telephone clinical interviews. Criteria for study inclusion were: (1) A recent documented clinical diagnosis of SLE given by an expert rheumatologist and/or clinical immunologist and verified via an interview with a lupus specialist from the study team. (2) Age 18 years or older (3) Hebrew speakers (4) Physical ability to attend MBSR sessions (5) Psychological ability to practice mindfulness (no serious cognitive impairments/psychosis/active suicidality/substance abuse) (6) No concurrent participation in another clinical study. *All SLE patients were diagnosed according to the American College of Rheumatology (ACR) criteria, by a lupus specialist. Upon inclusion in the study SLE diagnosis was re-confirmed* via *interviewing the patient and reviewing the medical chart for clinical and serological criteria of SLE.* 26 patients met eligibility criteria. All participants provided informed consent prior to their inclusion in the study. Patients were randomly assigned to either an MBSR group (*n* = 15) or a wait-list control group (WL; *n* = 11). Patients randomized to WL control group received no active treatment during their 10-weeks waiting period, at the end of which they received MBSR. Patients from the WL who chose to participate in the intervention after their waiting period (i.e., “delayed MBSR”; *n* = 5) were measured at the end of the intervention and their data were analyzed together with the original MBSR group (i.e., the “combined MBSR” group; *n* = 20).

The study received a Helsinki Committee approval by the ethics committee at Sheba Medical Center (No. 3296-16-SMC).

A detailed description of study procedures can be found in Figure 1 (PRISMA chart).

Table 1 presents the background characteristics of participants according to assignment group. Due to the wide heterogeneity of symptom type and severity in SLE, we also show levels of all SLE symptoms that are included in the SLAQ (Systemic Lupus Activity Questionnaire) at baseline, in Table 2.

### 2.2. Randomization and Measurement

Patients were randomized using the SPSS 23.0 statistical software package randomization algorithm. Patients in the WL were blind to the fact that they were allocated to the control group, as they were told that they were waiting for the next group opening. The therapists who conducted the treatment were blind to pre- and post-treatment evaluation data. Following randomization, paired-sample t-tests and Chi-square/Fisher tests were performed comparing between the WL and MBSR groups. No significant differences were found in terms of background characteristics or in any of the outcome measures. Finally, patients were asked about medical monitoring and treatment. All patients reported undergoing medical monitoring by a primary care physician or a rheumatologist and 22 patients were treated with Hydroxychloroquine; thirteen of these were from the MBSR group (86.67%) and nine were from the WL group (81.82%).

We collected data at three different stages: at baseline (before the first treatment session), at the end of the intervention (immediately after the final treatment session) and a follow-up assessment six months after treatment. The follow up assessment was conducted for the MBSR group only. Additionally, a sub sample of 12 patients took part in an in-depth qualitative interview, examining patients’ subjective experience of the intervention (see more details below).

We employed an (ITT) approach for our analysis, i.e., all randomized participants were contacted post-treatment to fill out questionnaires, regardless of whether they attended the entire therapy or dropped out. This was meant to provide a complete, reliable, and externally valid picture of our patient population.

### 2.3. Course Attendance

Patients showed relatively high attendance rates, with only one participant requesting to discontinue therapy. On average, patients in the MBSR group (*n* = 15) attended eight and a half out of eleven sessions. Six participants (40%) attended ten sessions, five participants (33.3%) attended nine sessions, three participants (20.1%) attended between five to eight sessions, and one participant (6.7%) attended three sessions and dropped out of treatment.

### 2.4. An Adapted MBSR Protocol for SLE

In order to create an adapted treatment protocol, more suited to the unique characteristics and difficulties of SLE patients, we used the standard MBSR 8-week format (University of Massachusetts, 1979) as the basis of the intervention, and then added modifications. The standard MBSR protocol was adapted in several ways:

1. The intervention was extended into a 10-week program with 11 sessions, which included shortened 2-h weekly sessions, due to potential physical difficulties for SLE patients, such as prolonged sitting. This extension enabled a significant expansion of the psycho-education section at the beginning of the intervention, where explanations about mindfulness, stress, and SLE were presented. The full-day retreat was also shortened into a 3-h session that took place after the eighth week.

2. The protocol included shorter, more carefully paced exercises, which gradually progressed from “easy” to “hard”. This was intended to facilitate SLE patients’ encounter with their body and physical sensations. An emphasis was placed on the concept of “pacing”, where patients were encouraged to strategize how much energy to exert according to which activities were planned for the day; “pacing” would prevent patients from being overly active when energy is available and feeling depleted by the end.

3. Therapists were instructed to pay special attention to SLE-related themes (e.g., pain and fatigue) throughout the sessions and practice, and to translate the general components of generic MBSR intervention into specific components relevant to SLE patients. For example, the issue of automatic thinking in the generic protocol would be translated into specific discourse about automatic pain-related thoughts and pain catastrophizing.

4. Therapists were instructed to specifically target maladaptive avoidance behaviors. Such behaviors are common among SLE patients, aimed to reduce physical efforts and movement and to prevent further diseases or infections.

5. Compassion, which is a vital component of MBSR, was emphasized to encourage a more flexible and accepting mindset towards the disease and the self. Further emphasis was placed in the protocol on decentering, which reflects the ability to observe one’s thoughts and feelings as transient, somewhat external mental states, as opposed to reflections of the self that are necessarily “true” and stable.

The amended protocol was carefully based on the standard components of the MBSR program, which were all included in the intervention (e.g., body scan, focus on breathing, mindfulness in daily life—see above). Home practice assignments included audio recorded mindfulness exercises (e.g., body scan, sitting mindfulness exercises, breathing exercises) in the therapist’s own voice. Daily home practice ranged from 5–30 min in duration, and patients were encouraged to exercise as much as possible.

The intervention was led by a licensed clinical psychologist who is an experienced MBSR teacher. Before commencing the complete RCT reported here, the study team held one pilot group which supported the feasibility of MBSR among SLE patients [11].

### 2.5. Measures

The systemic Lupus Activity Questionnaire (SLAQ) [26] is a 24-item questionnaire with a scoring range of 0–44. It also includes a single item (0–10) rating scale for SLE activity, and a rating scale for SLE flares (from no flare to severe flare). In the current study, SLAQ demonstrated excellent internal consistency, with a Cronbach’s alpha of 0.90.

The World Health Organization Quality of Life Questionnaire-Brief Version (WHOQOL-BREF) [27] was developed based on the original 100-item WHQOL questionnaire. It is a 26-item scale, with each item rated on a scale from 1 to 5. It covers four Quality of Life domains: physical, psychological, social, and environmental, additional to a global Quality of Life item. Cronbach’s alpha was very high, at 0.94.

The Patient Health Questionnaire-9 (PHQ-9) [28] is a commonly used self-reporting measure for symptoms of major depressive disorder. The scale scores each of the nine DSM-5 depression criteria from “0” (not at all) to “3” (nearly every day). It can yield either a continuous score, or a probable major depressive disorder diagnosis using a cut-off of 10. Cronbach’s alpha of the PHQ-9 in this study was very good, at 0.87.

Psychological Inflexibility in Pain Scale (PIPS) [29] is a 12-item measure based on a Likert scale of 1 to 7 per item, developed to assess psychological inflexibility towards pain (i.e., one’s ability or inability to manage pain in a flexible manner, not avoiding it altogether while at the same time not being “flooded” by it). The subscales measure avoidance (eight items) and cognitive fusion (four items). The total score ranges from 12 to 84. PIPS showed high reliability with Cronbach’s alpha of 0.91.

In order to measure SLE-related shame and identity, two single items assessed the degree to which the patient felt: (1) shame regarding her/his SLE; and (2) the degree to which SLE affects one’s personal identity. Both questions were assessed using a Visual Analog Scale (VAS), rated between 0 (minimum) to 100 (maximum), in order to allow for a wide spectrum of responses regarding these issues. The VAS has been used in numerous studies and randomized control trials in order to assess various emotional and physical symptoms among individuals suffering from chronic illnesses [30,31].

### 2.6. Qualitative Interviews

Out of the 26 participants, a sub-sample of 12 patients underwent a semi-structured qualitative interview regarding their subjective experience in therapy, at post treatment (60% of patients from the MBSR group; *n* = 9; 60% of patients from the delayed MBSR group; *n* = 3). The interviews were conducted on average 15 days following treatment (SD = 3.97). The average duration of the interviews was 48.25 min (SD = 4.76; range 44–55 min). The interviews were meant to explore participants’ subjective experiences, in their own words, in order to acquire a deeper understanding of changes in the psychological and physical aspects. The interviews were semi-structured, leaving room for both guideline-based questions as well as a more explorative approach. They started with an open question (e.g., “What comes to mind when you think about the 10-week mindfulness program?”) and then focused on the group (e.g., “How did you feel among the group?”), the experience with the instructor, intervention content (“how did you experience the exercises in the sessions?”) and potential difficulties with the exercises. The next part of the interview focused on the disease and the extent towhich the intervention was relevant to its physical aspects (e.g., “Do you feel changes in your coping with SLE?”), and psychological aspects (e.g., “What changes, if any, have you noticed in your mental and psychological state since joining the intervention?”).

Analysis was based on Renner and Taylor-Powell’s [32] systemic approach for analyzing qualitative data, which requires categorizing the themes which emerge from the text into primary themes and sub-themes, as well as classifying the themes according to their frequency, allowing for a better understanding of not only which of the themes emerge, but also how often they do so.

In order to attain interrater reliability, two independent raters analyzed three random transcripts of interviews, categorizing the texts into main themes. Cohen’s Kappa was 0.830, which indicated excellent inter-rater agreement [33].

## 3. Results

We will report here three types of results: (1) differences the MBSR and WL groups in changes over time (on the various outcome measures). (2) the long-term effects of MBSR, as seen in the follow-up measurement of the treatment group and (3) within-group effects in the larger MBSR group, which includes both the MBSR and delayed MBSR (e.g., post-WL) group.

### 3.1. Pre- to Post-Treatment Differences between the MBSR and WL Groups

In order to examine changes in outcome measures among MBSR participants compared to WL controls, we conducted a series of repeated measures multivariate analysis of covariance (MANCOVA) for Group X Time interactions, adjusting for age as a covariate (due to its important role in SLE, as well as due to the wide age range in our sample). For the sake of brevity, we report here only measures where at least some significant effects were found.

For SLE-related Shame and Illness Identity, the repeated measures MANCOVA yielded a significant effect for Time (F(1, 22) = 5.72, *p* < 0.01), a non-significant effect for Group (F(1, 22) = 2.85, n.s), and a significant Group X Time interaction (F(1, 22) = 3.80, *p* < 0.05). Subsequent univariate analysis for Shame yielded a significant interaction for Group X Time, while that for Illness Identity was marginally significant. The multivariate analysis for Quality of Life showed a significant effect for Time (F(1, 24) = 2.77, *p* < 0.05), but not for Group (F(1, 24) = 0.83, n.s) or for the Group X Time interaction (F(1, 24) =3.10, n.s). However, due to our small sample size, as well as to the clear directionality of our hypotheses (MBSR > WL), we went on to examine the univariate interaction effects. In the subsequent univariate analysis of variance for Quality of Life, a significant interaction was found for the General subscale and for the Environmental subscale. Repeated measures univariate effect for the Physical Quality of Life subscale was marginally significant. Univariate tests for the other subscales did not reach significance. For Psychological Inflexibility in Pain, the MANCOVA was non-significant for Group X Time interaction (F(1, 24) = 2.39, n.s) and for Time (F(1, 24) = 2.17, n.s). The MANCOVA for Group was significant (F(1, 24) = 3.68, *p* < 0.05). As above, we went on to examine univariate interactions, where a significant effect was detected for the Fusion with Pain subscale. Univariate tests for Avoidance of Pain did not reach significance. Table 3 presents univariate effects and descriptive statistics for between-group comparisons, as well as for MBSR follow-up (discussed later).

Next, we conducted subsequent pairwise comparisons for all significant Group X Time effects, in order to establish the source of the interaction. Comparison of the MBSR versus WL groups in Physical QOL, Shame and Fusion, revealed significant improvements in the MBSR group (Physical Quality of Life: i − j = 1.35, SE = 0.55, *p* = 0.02; Shame: i − j = −17.32, SE = 5.53, *p* = 0.005; Fusion: i − j = −3.48, SE = 1.189, *p* = 0.008), and no significant change among WL controls group (Physical Quality of Life: i − j = −0.33, SE = 0.64, *p* = 0.61, Shame: i − j = 5.9, SE = 6.49, *p* = 0.37; Fusion: i − j = 0.38, SE = 1.4, *p* = 0.79). General and Environmental QOL showed non-significant improvements in the MBSR group (General QOL: i − j = 0.79, SE = 0.62, *p* = 0.22; Environmental QOL: i − j = 0.65, SE = 0.41, *p* = 0.13) and non-significant worsening in the WL group (General QOL: i − j = −1.25, SE = 0.72, *p* = 0.1; Environmental QOL: i − j = −0.75, SE = 0.48, *p* = 0.13). Thus, the interaction seemed to have stemmed from the general difference in trend in both groups (improvements/worsening). Figure 2 presents effects of all significant interactions.

### 3.2. Long-Term Effectiveness of MBSR

To assess the long-term effect of MBSR intervention, we conducted univariate and multivariate analysis for follow up assessment six months after the last MBSR session. We base this analysis on the original MBSR group alone (*n* = 15), as this group was the only one with three assessments. We generally expected that the contrasts between pre- and post-treatment would be significant, while those between post-treatment and the 6-month follow-up will be non-significant, showing a stability of identified treatment effects over time.

A repeated measures within-group multivariate analysis of variance was conducted, with three time points-pre-MBSR, post-MBSR and six months follow-up. The within-group MANOVA for SLE-related Shame and Illness Identity was significant (F(2, 13) = 3.99, *p* < 0.05), while the MANOVA for Psychological Inflexibility in Pain was marginally significant (F(2, 13) = 3.12, *p* = 0.066). Subsequently, univariate effects were calculated for all variables, in order to detect the source of the effect (see Table 3).

As can be seen in Table 3, univariate analysis was significant for SLE-related Shame and Illness Identity. Subsequent contrasts for Shame showed a significant effect between pre- and post- intervention assessments (*p* < 0.05, Cohen’s d = 0.64) and a non-significant effect between post-intervention and follow-up (*p* = 0.13), indicating effect stability. Illness Identity showed a significant effect between pre- and post-intervention assessments (*p* < 0.01, Cohen’s d = 0.83), and a significant increase again between post- intervention and follow- up (*p* < 0.01), showing that the benefit of MBSR was relatively short-lived.

Next, a univariate significant effect was found for the Fusion with Pain subscale. There was no significant effect for Avoidance of Pain. For Fusion, a significant improvement was found between pre- and post-intervention assessments (*p* < 0.01, Cohen’s = 0.98), while a non-significant effect was found between post-intervention and follow-up (*p* = 0.83), indicating effect stability. Figure 3 presents the follow-up effects of Fusion with Pain and SLE-related Shame.

Finally, we set out to examine changes in Depression rates across time. We calculated the rates of participants who met the PHQ-9 cutoff (sum > 10) for a probable diagnosis of major depressive disorder (MDD). As can be seen in Figure 4, the analysis showed that in the original MBSR group, 40% of participants met the MDD cutoff before treatment, followed by 33% post-treatment, and 28.5% at the 6-months follow-up. Thus, rates of MDD declined consistently over time.

### 3.3. Pre- to Post-Intervention Changes within the Combined MBSR Group (n = 20)

In subsequent analysis, we included all participants who underwent MBSR, combining the original MBSR group with the “delayed MBSR” group (i.e., WL participants who subsequently went on to complete MBSR, and were measured pre- and post-treatment). This enabled a closer examination of all those undergoing MBSR, relaying on a larger sample (*n* = 20) to identify treatment effects.

First, repeated-measures analysis of variance for SLE symptoms (based on the SLAQ measure) showed a significant treatment effect. Next, a repeated measures multivariate analysis of variance showed a significant main effect for psychological inflexibility in pain (F = 9.36, *p* < 0.01), as well as for SLE-related Shame and Illness Identity (F = 5.10, *p* < 0.05). Subsequently, in order to examine what were the specific components that contributed to the observed changes, a univariate analysis was conducted, yielding significant treatment effects for the Fusion with Pain subscale, but not for Avoidance of Pain. Univariate analysis also showed a significant effect for SLE-related Shame and Illness Identity.

Table 4 presents main effects for the MBSR combined group (only measures with at least one significant effect are presented).

In order to examine more thoroughly the relationships between the change in SLE symptoms and the change in psychological outcomes following MBSR, we also calculated Pearson correlations between each SLE symptom (according to the SLAQ questionnaire) and psychological changes (e.g., Shame, Quality of Life, pain cognitions), by calculating the score at the end of the intervention minus the score at the beginning of the intervention for each measure. A correlation was then calculated between each difference score.

The change in SLE-related Shame was related to the changes in fever (*r* = 0.70, *p* < 0.01), presence of dark, blue or purple spots on the skin (*r* = 0.65, *p* < 0.01), rash or feeling sick after sun exposure (*r* = 0.48, *p* < 0.05), bald patches on scalp, or clumps of hair on pillow (r = 0.57, *p* < 0.01), swollen glands in the neck (*r* = 0.48, *p* < 0.05), shortness of breath (*r* = 0.62, *p* < 0.01) and forgetfulness (*r* = 0.52, *p* < 0.05). The change in General Quality of Life was related to the change in forgetfulness (*r* = −0.44, *p* < 0.05). The change in Psychological Inflexibility in Pain was related to the changes in fatigue (*r* = 0.49, *p* < 0.05), headaches (*r* = 0.51, *p* < 0.05) and muscle pain (*r* = 0.49, *p* < 0.05). The change in Pain Avoidance was related to the change in bald patches on scalp, or clumps of hair on pillow (*r* = 0.54, *p* < 0.05), muscle pain (*r* = 0.50, *p* < 0.05), and muscle weakness (*r* = 0.45, *p* < 0.05).

### 3.4. Qualitative Analysis of In-Depth Interviews

As noted above, 12 qualitative interviews were analyzed to identify the major recurring themes related to participants’ experience during and following MBSR. Analysis of participants’ transcripts generated five primary themes: (1) changes related to mindfulness; (2) stress reduction; (3) improvement in general physical functioning; (4) changes in illness identity and illness perception; and (5) the group as a mechanism of psychological change.


*
Changes related to mindfulness
*


Throughout the 12 analyzed interviews, participants commonly referred to changes related to their ability to be mindful. Mindfulness is a multi-faceted concept, and accordingly, we have identified several sub-themes in the interviews:

(a)Non-reactivity to inner experiences: Ten out of twelve participants referred to their in-creased ability, following MBSR, to be less reactive or impulsive in the face of distress. As one participant noted: *“…I am very impulsive… and it became worse for me… I feel like this workshop has helped me. I don’t really count ‘til ten, but it did somewhat help me to delay my response and really choose it before I react one way or the other”.*(b)Observing one’s negative feelings and physical sensations: 7 of 12 participants have referred to their increased ability to focus their attention on what they feel or experience in a given moment, thus being able to better understand and regulate their physical and psychological experiences. The following quote exemplifies this notion: *”(during meditation) I can sit for 20 min, even 30 min, and I don’t think about when it’s going to end… I go back to feeling my breath or my sensations, and then I can remain there with a good feeling. I feel I can actually benefit from that time”*.(c)Non-judgement towards one’s negative emotions or physical sensations: six of twelve participants referred to the fact that MBSR helped them feel less self-critical towards themselves as SLE patients, as well as allowing them to feel more self-compassionate about their symptoms and distress. This participant described what is happening to her during SLE flares: *“I’ve learned how to say ‘OK, so today I wasn’t able to do this or that-I am not going to give myself a hard time about it. If it doesn’t work, so it doesn’t work, it’s all good. It gives me some kind of… ease”.*


*
Stress reduction
*


A highly prominent theme, which emerged from nine of the interviews, concerned the changes participants experienced in terms of their stress-related cognitions and reactions. Participants reported a reduction in stress following the intervention, stemming from the exercises they learned, as well as from their ability to be more mindful about their physical and emotional sensations. For example, one participant noted: *“When I feel my heart pounding, I listen to my pulse and it’s simply calming it down. I can feel the pressure dropping, the lungs expanding, and I can breathe deeper”*. The improved mental skills mentioned in the interviews included participants’ ability to distance themselves from distressing experiences, and leave stressful thoughts aside, while placing distressing events within a more acceptable, manageable, and less catastrophic perspective.


*
Improvement in general physical functioning
*


In line with what was found in the quantitative section of this paper, interviews also showed an improvement in physical functioning in everyday life. Five participants described having formed a new and adaptive attitude towards their body, which enabled them to function more freely and to feel less limited by their symptoms. In the words of one participant: *“When I have to brace myself and go to work, I listen to the body scan exercise and it helps me to physically pick up my body”*. In another instance, a participant noted: *“My body told me what it needed… There was a connection between what I was going through and what my body was going through”*.


*
Changes in Illness identity and perception
*


In line with our quantitative findings, six out of twelve interviewees reported a positive change in their identity as SLE patients following the intervention. Participants described how they developed a healthier and less stigmatic attitude towards their illness, and its accompanying symptoms. Participants also mentioned how the intervention has helped them to be less defined by their SLE, for example: *“I want to hope that my Lupus would stay calm, but if it will decide to flare, it would be a bit easier to accept the episode… but I don’t let it define me”*.


*
The group as a mechanism of psychological change
*


Another theme strongly emerging from the interviews concerned the therapy group and its positive effect on the individual. Six participants stated that the intervention set up a unique opportunity to meet other SLE patients who were suffering from similar physical and psychological symptoms: *“It felt good to be part of this ‘togetherness’. I met other strong women that have been coping with life and death for many years, and still continue to live their lives... they are mothers that gave birth with Lupus, and this gave me a lot of strength, and maybe even hope”.* The group was portrayed by participants as very important in terms of solidarity. Group members mirrored to each other their ability to cope with SLE, thus creating a strong reciprocal support system for participants.

## 4. Discussion

We conducted a randomized controlled trial to examine the effects of MBSR group therapy for SLE, employing both self-reporting measures and qualitative in-depth interviews. For this study, we developed an adapted MBSR protocol, specifically suited to meet the needs and symptoms of SLE patients. The results showed the positive effects of MBSR on SLE self-reported symptoms, Quality of Life, Depression, SLE-related Shame and Illness Identity, and Psychological Inflexibility in Pain. Importantly, some effects remained stable over a 6-month follow-up period.

Surprisingly, mindfulness-based interventions have been very rarely studied in the context of SLE, despite a growing body of evidence showing their effectiveness on rheumatic [34], autoimmune [35] and pain [36] conditions. In addition, to the best of our knowledge, ours is the first mixed-methods RCT of MBSR, as well as the only MBSR study of SLE to include a variety of psychological and SLE-related measures. The effectiveness of MBSR shown here may be attributed to several factors. Firstly, many studies have indicated the major role of mind-body connections in SLE. This is perhaps most apparent in the strong associations between psychological stress and SLE disease activity [4]. Thus, an intervention with proven efficacy in reducing stress could be expected to alleviate the distress of SLE patients. Secondly, and perhaps more importantly, MBSR inherently targets various aspects which have been shown to be at the heart of SLE-related distress, including impaired emotion regulation, negative mood, and difficulties in identifying and acknowledging various feelings and emotions (i.e., alexithymia; [37]). Finally, some of the core symptoms of SLE, most notably pain and chronic physical discomfort, have been shown to improve via MBSR in previous studies [38].

Furthermore, our findings reveal the unique role MBSR may have in alleviating feelings of shame and negative illness perception among individuals with SLE. These treatment effects were found in both quantitative and qualitative analysis conducted in this study. Shame is a common emotional feature of SLE. It may stem from both direct physical symptoms (e.g., facial rashes, alopecia, weight gain) as well as from the broader experience of living with a chronic disease that undermines daily functioning and affects femininity [39,40]. The decline in SLE-related shame remained stable at the 6-month follow-up assessment, thus pointing to the unique potential of MBSR in this area. As for the perception of illness, studies have shown that SLE, being a systemic, chronic condition, often exerts a massive influence on patients’ lives, and may be experienced as a major focal point in their identity. Schattner and colleagues [41] has showed that SLE patients may perceive their illness as all-encompassing and extremely powerful, “…threatening the ties between body, mind, everyday life, and relationships”. MBSR’s strong emphasis on cultivating a non-judgmental stance, as well as its proved ability to promote self-compassion [42], may thus greatly contribute to SLE patients’ well-being. However, we should note that while illness identity did improve between pre- and post-treatment assessments, this improvement did not seem to last when measured again at six months post-treatment. This may indicate that some aspects of SLE-related distress may require more long-term psychotherapeutic work and maintenance. One’s perception of his/her identity as a person coping with a chronic illness is often multi-layered and complex. Further studies are needed to assess whether mindfulness-based therapy can serve as an add-on component used to augment other forms of interventions (e.g., cognitive-behavioral, psychodynamic) for SLE.

Another specific effect worth noting is related to pain cognitions. Pain is a highly distressing symptom for some SLE patients [43]. It is a complex sensation, involving both objective, physiological aspects, and subjective, psychological ones. Importantly, these two realms of pain were often found to affect each other, creating both positive and negative feedback loops [44]. In our study, we have shown that MBSR may cultivate a more flexible attitude towards physical pain among SLE patients. This was mostly found specifically for fusion with pain (i.e., feeling that one’s pain is taking over one’s body, and even life), which not only declined following treatment, but also showed stable gains after six months. Due to pain’s far-reaching implications in terms of quality of life and daily functioning, this effect on SLE patients is of unique importance.

Several other effects are noteworthy. Major depression levels (i.e., percentage of those above/below clinical cutoff) showed a steady decline across the three assessments. This improvement is important, given the well-established association between SLE and negative mood [45]. Our qualitative analysis added two interesting effects. First, as would be expected from MBSR, the intervention improved patients’ ability to be mindful on several aspects. In general, they reported an increased ability to observe and regulate emotions and sensations, while adopting a non-reactive and non-judgmental stance. These results were in line with another theme stemming from the interviews, which indicated decreased stress levels. Patients seemed to use MBSR to feel calmer, and to decrease psychological tension.

Interestingly, the reduction shown in this study in SLE self-reported symptoms was associated with an improvement in mental health outcomes (Quality of Life, SLE-related shame and pain cognitions). We believe that this further supports the mind-body connection in SLE, which has already been established in both quantitative [46] and qualitative [47] studies. Thus, when one’s physical symptoms improve, so too do aspects of psychological distress, and vice versa. MBSR, being an inherently mind-body approach, targets both physical and psychological aspects in parallel.

This study has several limitations, most notably a modest sample size, and a reliance of self-reporting measures, which may be prone to memory or reporting bias. Thus, our findings should be considered with the adequate level of caution, as they warrant further research for support. Future studies are encouraged to employ larger samples, objective measures of physical and psychological distress, and more assessment points. Nonetheless, our findings have clinical and methodological importance. They point to the unique potential of MBSR to alleviate the distress of SLE patients, using a group intervention characterized by high cost-effective value (i.e., group setting, short time period). Methodologically, we have shown here, for the first time ever, how qualitative and quantitative methods could complement each other to assess SLE patients’ response to MBSR. Finally, all of the effects discussed here should be examined in light of our attempt to offer SLE patients an adapted, disease-specific MBSR protocol. In this era of personalized medicine [48], there is a growing attempt to adapt psychotherapy protocols in general, and MBSR protocols specifically [49], to the needs and characteristics of unique patient populations. We hope that, with more research to follow the present study, this protocol can be used in multiple settings to treat individuals with SLE.

## Figures and Tables

**Figure 1 jcm-10-04450-f001:**
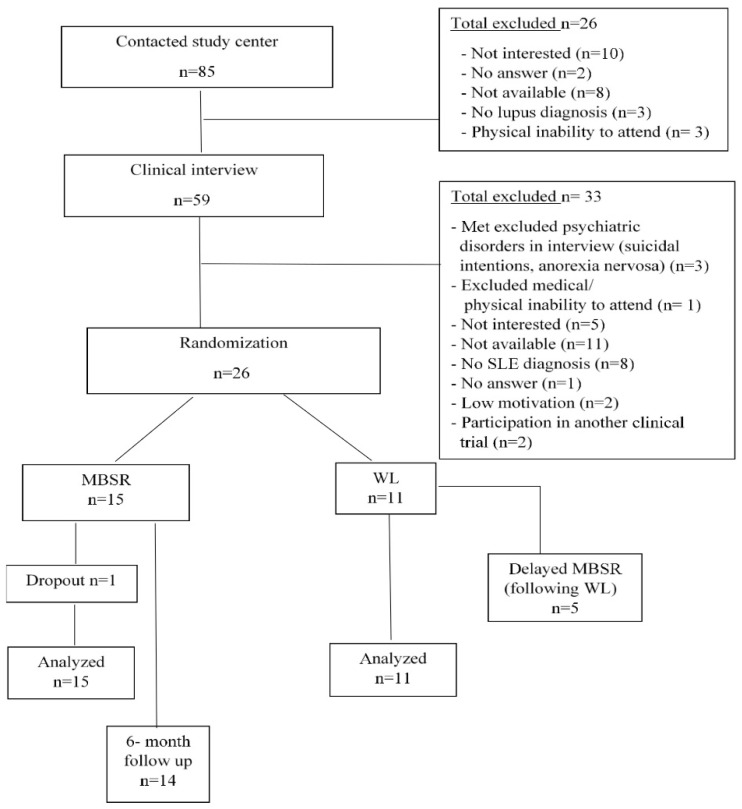
PRISMA flow- chart for screening and allocation process.

**Figure 2 jcm-10-04450-f002:**
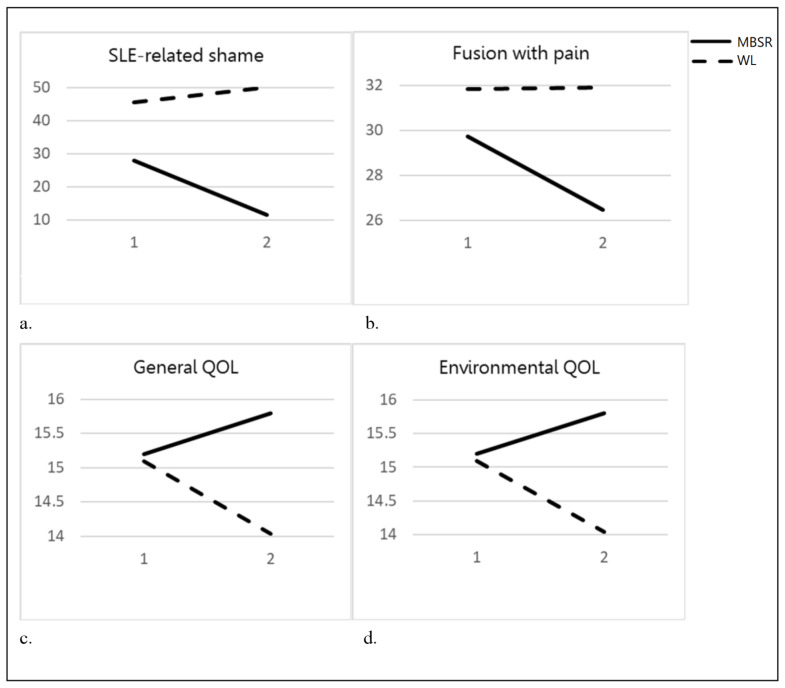
Pre- Intervention to Post-Intervention Differences between the MBSR and WL. Note: Pre-intervention (1); post-intervention (2). (**a**) Changes in cognitive Fusion among MBSR group and WL; (**b**) Changes in SLE-related Shame among MBSR group and WL; (**c**) Changes in General Quality of Life among MBSR group and WL; (**d**) Changes in Environmental Quality of Life among MBSR group and WL.

**Figure 3 jcm-10-04450-f003:**
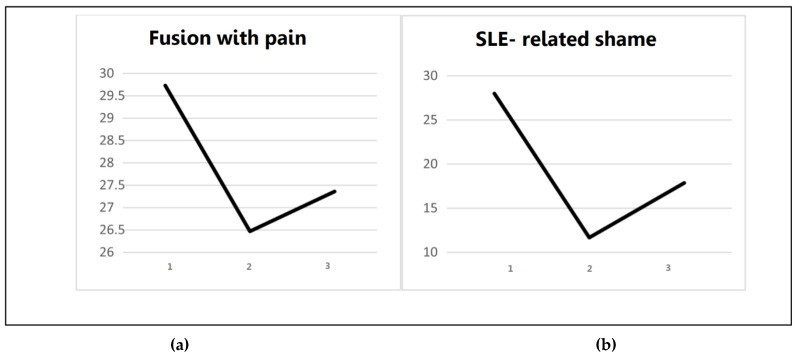
Long- Term effects of SLE- related Shame and Cognitive Fusion with Pain. Note: Measurements at pre- intervention (1); post-intervention (2); 6-month follow-up (3). (**a**) Follow- up effect of SLE- related shame among MBSR group; (**b**) Follow-up effect of Fusion with pain among MBSR group.

**Figure 4 jcm-10-04450-f004:**
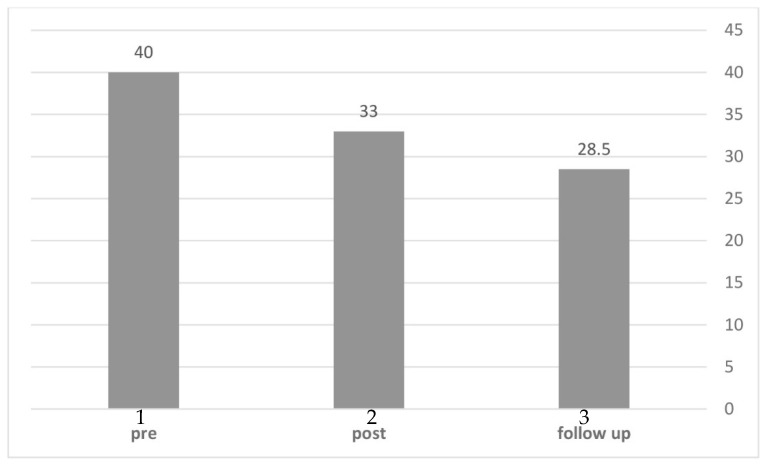
Rates (%) of Major Depressive Disorder in pre-, post- and follow-up assessments (MBSR group). Note: Pre-intervention (1); post-intervention (2); 6-month follow- up (3).

**Table 1 jcm-10-04450-t001:** Baseline characteristics according to group.

Characteristic	MBSR Group *n* = 15	Wait-List Controls *n* = 11
Gender	Male (*n* = 1, 6.67%) Female (*n* = 14, 93.33%)	Male (*n* =2, 18.18%) Female (*n* = 9, 81.82%)
Age (in years, mean, SD)	43.93 (11.78)	38.45 (10.56)
Duration of disease (in years, mean (SD)	11.00 (10.06)	11.73 (8.73)
Disease activity during the past month	4.40 (2.47)	3.91 (2.63)
Hospitalized due to SLE complications	64% (*n* = 9)	81% (*n* = 9)
Treated by rheumatologist	100%	100%
Pharmacological treatment	100%	100%
Family status-%		
Married/ In a relationship	10 (66.67%)	7(63.63%)
Single/Divorced	5 (33.33%)	4 (36.36%)
Education level-%		
12th grade or less	4 (26.66%)	3 (27.27%)
Currently student/Academic degree	11 (73.33%)	8 (72.72%)
Income		
Under average	3 (21.43%)	4 (36.36%)
Average and above	11 (78.57%)	7 (63.64%)

**Table 2 jcm-10-04450-t002:** Baseline SLE Symptoms according to the SLAQ (on a Likert scale of 0 (no problem) to 3 (Severe).

SLE Symptom	Baseline Severity Level of Symptom Mean (SD)
Flares	1.27 (0.83)
Lost weight	0.73 (1.08)
Fatigue	1.69 (1.23)
Fevers (>101 °F, 38.5 °C) taken by a thermometer	0.38 (0.64)
Sores in mouth or nose	0.73 (1.00)
Rash on cheeks (butterfly shaped)	0.73 (0.78)
Other rash	0.85 (0.97)
Dark blue or purple spots on the skin	0.96 (0.96)
Rash or feeling sick after sun exposure	1.04 (1.00)
Bald patches on scalp, or clumps of hair on pillow	0.81 (0.94)
Swollen glands (nodes) in the neck	0.58 (0.81)
Shortness of breath	0.69 (0.84)
Chest pain with a deep breath	0.92 (0.93)
Fingers or toes turning dead white or very pale in the cold (Raynaud’s)	1.12 (1.11)
Stomach or belly pain	0.92 (0.98)
Persistent numbness or tingling in arms or legs	1.23 (1.03)
Seizures	0.65 (0.85)
Stroke	0.50 (0.86)
Forgetfulness	1.35 (1.02)
Feeling depressed	1.38 (1.10)
Unusual headaches	1.00 (0.89)
Muscle pain	1.35 (1.26)
Muscle weakness	1.19 (1.10)
Pain or stiffness in joints	1.23 (1.27)
Swelling in joints	0.53 (0.76)

**Table 3 jcm-10-04450-t003:** MBSR vs. WL and MBSR follow-up effects and descriptive statistics.

Outcome	Scale	MBSR (*n* = 15)			WL (*n* = 11)		Treatment Effects				6-Month Follow-Up (MBSR Alone)
		Pre	Post	6-Month Follow-Up	Pre	Post	F Time	F Group	F Univariate Time X Group	Time X Group Cohen’s d	F
		Mean									
		(SD)									
**Health-Related Quality of life**	General	12.93 (2.60)	13.47 (2.56)	12.71 (2.02)	11.82 (2.89)	10.9 (3.14)	5.96 *	3.91	4.49 *	0.52	0.62
	Environmental	15.20 (2.46)	15.80 (1.98)	15.39 (2.40)	15.09 (2.69)	14.41 (3.05)	0.46	0.73	4.75 *	0.49	0.27
	Physical	14.13 (3.31)	12.95 (2.82)	13.22 (3.00)	11.58 (4.22)	11.48 (3.77)	5.11 *	2.16	3.90~ (*p* = 0.06)	−0.28	0.40
	Social	13.51 (3.01)	13.33 (13.15)	13.71 (3.16)	12.24 (4.08)	12.97 (3.32)	0.05	0.17	0.96	−0.25	0.84
	Psychological	14.76 (3.16)	15.11 (2.74)	14.38 (2.92)	13.09 (3.54)	12.67 (3.33)	0.01	2.36	1.25	0.22	0.38
**Psychological Inflexibility in Pain**	Fusion with pain	29.73 (4.92)	26.47 (5.97)	27.36 (6.96)	31.82 (5.79)	31.91 (3.75)	2.73	2.53	4.32 *	−0.61	7.69 *
	Avoidance of pain	31.31 (11.98)	29.27 (6.92)	30.14 (7.40)	44.36 (14.75)	43.73 (14.4)	1.40	7.38 *	0.39	−0.10	0.91
**SLE-related Shame and Illness Identity**	Shame	28.00 (33.42)	11.67 (16.00)	17.86 (24.86)	45.45 (35.88)	50.00 (37.41)	2.04	5.20 *	6.78 *	−0.59	4.99 *
	Illness identity	33.33 (25.82)	18.33 (19.24)	28.57 (24.45)	56.00 (33.73)	50.00 (33.66)	11.83 **	5.05 *	3.36~ (*p* = 0.08)	−0.30	7.41 *

* *p* < 0.05; ** *p* < 0.01; ~marginally significant.

**Table 4 jcm-10-04450-t004:** Pre- to post-intervention changes in the combined MBSR group.

Combined MBSR (*n* = 20)				F	Cohen’s d
		Pre	Post		
SLE symptoms	SLE (SLAQ)	26.05	20.35	5.32 *	**0.52**
		(16.5)	(10.64)		
SLE-related Shame and Illness Identity	Shame	35.91	27.5	4.79 *	0.33
		(36.6)	(34.56)		
	Illness Identity	43.64	32.05	10.73 **	0.57
		(30.79)	(31.12)		
Psychological Inflexibility in Pain	Fusion	29.9	26.55	18.18 ***	0.95
		(4.48)	(5.8)		
	Avoidance	33.75	32.4	0.63	0.17
		(13.8)	(11.07)		

Note: * *p* < 0.05; ** *p* < 0.01; *** *p* < 0.001.

## Data Availability

The data presented in this study are available on request from the corresponding author. The data are not publicly available due to privacy and confidentiality.

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
