# Peer review of "Mindfulness-Based Stress Reduction for Systemic Lupus Erythematosus: A Mixed-Methods Pilot Randomized Controlled Trial of an Adapted Protocol"

_jcm, 2021, doi:10.3390/jcm10194450_

Round 1

Reviewer 1 Report

The Authors analysed the potential of a Mindfulness-Based Stress Reduction (MBSR) approach to improve the psychological distress of SLE patients. The topic is of interest to this area of research since the psychological burden affecting SLE patients should not be disregarded.

Major comments:

  • Methods: The small number of patients limits the value of the analysis.

  • SLE an extremely heterogeneous disease and patients might experience complete different clinical manifestations with different degrees of severity. It would be of interest to know what the clinical manifestations of the patients are, and to investigate how much some clinical manifestations, such as coetaneous involvement, are associated with a psychological burden through MBSR. Baseline conditions and follow-up disease status are also relevant, as patients’ feeling might be favourably influenced by remission of disease or negatively influenced by a disease flare. Furthermore, differences in clinical manifestations at baseline and follow-up in the treatment group and control group might be relevant.

  • Patients: was diagnosis of SLE confirmed on clinical ground? Of course only a telephone interview cannot be thought acceptable?

Minor comments:

  • Patients: while SLE impacts the QoL and Psychological status of the patients, whether the patients included in the study suffered from underlying depression, anxiety or other conditions that might impact on the results of the study it is not specified.

Author Response

Below please find a response for each and every comment (the reply appears below the comment in Italics). Inside the revised text, all new additions appear in track changes as well as marked in yellow.

Reviewer 1:

Major comments:

  • Methods: The small number of patients limits the value of the analysis.
  • Response: We agree with the reviewer that our sample size is quite modest. However, it should be noted that an N=26 is by no means rare in psychotherapy studies. Many RCTs with a similar or lower N are frequently published in very good journals. This is also specifically true for studies examining psychological interventions for autoimmune or rheumatic diseases. Please see the following reviews:

Bernardy, K., Füber, N., Köllner, V., & Häuser, W. (2010). Efficacy of cognitive-behavioral therapies in fibromyalgia syndrome—a systematic review and metaanalysis of randomized controlled trials. The journal of Rheumatology37(10), 1991-2005.‏

Dissanayake, R. K., & Bertouch, J. V. (2010). Psychosocial interventions as adjunct therapy for patients with rheumatoid arthritis: a systematic review. International journal of rheumatic diseases13(4), 324-334.‏

In addition, please take into account that mindfulness studies in SLE are extremely rare, and thus an RCT of this size still sheds very important light on this under-studies field, even with a modest sample.

That being said, following the reviewer’s comment we have now further emphasized the preliminary nature of our findings, given the sample size (see abstract, as well as pages 3 and 19). We have also modified the paper’s title accordingly. WE also clearly note in the text the need for further studies, employing a larger sample size, as well as the need to consider our results with the proper degree of caution (page 19). Finally, we clearly note the sample size as a limitation in the limitations paragraph (page 19).

We also wish to state that, despite the modest sample size, the RCT shows promising  results based on highly accepted statistical methods. We hope this too attests to the validity of our analysis and study as a whole.

  • SLE is an extremely heterogeneous disease and patients might experience complete different clinical manifestations with different degrees of severity. It would be of interest to know what the clinical manifestations of the patients are, and to investigate how much some clinical manifestations, such as coetaneous involvement, are associated with a psychological burden through MBSR. Baseline conditions and follow-up disease status are also relevant, as patients’ feeling might be favourably influenced by remission of disease or negatively influenced by a disease flare. Furthermore, differences in clinical manifestations at baseline and follow-up in the treatment group and control group might be relevant.
  • Reply: Thank you for this very insightful comment. Indeed, we probably did not include sufficient information about SLE symptoms in the previous version, and we have now rectified this in light of the review. In our study, we have administered a comprehensive questionnaire assessing SLE symptoms, called the SLAQ. We have now done several things to give the reader a better sense of SLE symptomatology and its effect in our study:
  • On page 7, we have added the new Table 2, which shows baseline severity levels for all the specific SLE symptoms included in the SLAQ.
  • As was suggested by the reviewer, we have conducted new statistical analysis to show the association between changes in SLE symptoms following MBSR (i.e., how did the intervention affect symptoms) and mental health changes following MBSR (page 16). This new analysis presents an interesting picture, according to which positive changes in symptoms were associated with positive changes in mental health following the intervention, thus further supporting the mind-body connection in SLE. Since this was a completely new analysis, we have also now set the stage for this issue in the introduction (page 3), and we of course also discuss it in the discussion (page 19), with new references added.

  • Patients: was diagnosis of SLE confirmed on clinical ground? Of course only a telephone interview cannot be thought acceptable?
  • Reply: You are of course correct in your comment, and we realize we did not clarify this enough in our previous version. As we now note in the revised methods section (page 3), in this study SLE diagnosis was confirmed on clinical ground. All SLE patients were diagnosed according to the American College of Rheumatology (ACR) criteria, by a lupus specialist. Upon inclusion in the study SLE diagnosis was re-confirmed via interviewing the patient and reviewing the medical chart for clinical and serological criteria of SLE. As noted, this is now clearly stated in the manuscript itself.

Minor comments:

  • Patients: while SLE impacts the QoL and Psychological status of the patients, whether the patients included in the study suffered from underlying depression, anxiety or other conditions that might impact on the results of the study it is not specified.
  • Reply: We in fact asked participants about existing psychiatric diagnosis, but no one reported having such a diagnosis. As for depression levels – please see the rates reported by us in the relevant section of the results section. Although patients did not report having received a diagnosis of depression, we nonetheless found that a considerable number of patients began the study with a clinically significant level of depression (score of 10 and above on the PHQ-9 scale which we administered). This also appears in a figure.

Reviewer 2 Report

The article present methods for mindfulness-based stress reduction in SLE, methods that proved therapeutic potential for important aspects like pain, acceptance and quality of life. There are several limitations of the study presented, that were acknowledged in the paragraph dedicated to this. Overall, I find this research interesting and important for daily practice.

Author Response

Below please find a response for each and every comment (the reply appears below the comment in Italics). Inside the revised text, all new additions appear in track changes as well as marked in yellow.

Reviewer 2:

The article present methods for mindfulness-based stress reduction in SLE, methods that proved therapeutic potential for important aspects like pain, acceptance and quality of life. There are several limitations of the study presented, that were acknowledged in the paragraph dedicated to this. Overall, I find this research interesting and important for daily practice.

Reply: Thank you very much for this positive feedback.

Reviewer 3 Report

This is a very interesting manuscript reporting the effects of mindfulness in  cohort of SLE patients. Overall, the study is well conducted and written, the Methods section very accurate and the Results well presented.

I have a few minor comments:

  • Table I needs improvement. I understand the difficulties linked to the social media enrollment, but further information regarding the SLE (i.e. disease activity, ongoing or previous treatment) should be reported.
  • Was any patient taking anti-depressants or anxiolytic drugs before or after enrollment? Was it an exclusion criterion? Please, explain.

English is fine and only needs minor spell checks.

Author Response

Below please find a response for each and every comment (the reply appears below the comment in Italics). Inside the revised text, all new additions appear in track changes as well as marked in yellow.

Reviewer 3:

This is a very interesting manuscript reporting the effects of mindfulness in  cohort of SLE patients. Overall, the study is well conducted and written, the Methods section very accurate and the Results well presented.

I have a few minor comments:

  • Table I needs improvement. I understand the difficulties linked to the social media enrollment, but further information regarding the SLE (i.e. disease activity, ongoing or previous treatment) should be reported.
  • Reply: Thank you for this important comment. Following this comment, we have now revised Table 1, by adding more information about SLE: disease activity, hospitalization, pharmacological treatment, and more.
  • Was any patient taking anti-depressants or anxiolytic drugs before or after enrollment? Was it an exclusion criterion? Please, explain.
  • Reply: Thank you for this question. In fact, we did check for medicine use, and found that only 2 women (both from the WL group) reported having used antidepressants (one used Citalopram and the other used Sertraline). Due to the very small number of women (n=2) using these drugs, we of course had no way to analyze their effects. As for exclusion criteria: no, psychopharmacological drugs were not an exclusion criterion. This is common practice in many psychological RCTs. Asking a patient to get off medication for months would be unethical in most cases. Thus, many studies simply monitor this issue rather than exclude (just as we did here, thereby being able to provide info on the N=2 who took medication). Again, in our case the numbers were so small, this does not pose a major issue anyway.

  • English is fine and only needs minor spell checks.
  • Reply: The entire MS was re-checked and proof-read